# Explaining Image Classifiers by Counterfactual Generation

**Chun-Hao Chang, Elliot Creager, Anna Goldenberg, & David Duvenaud**
University of Toronto, Vector Institute
{kingsley,creager,duvenaud}@cs.toronto.edu
anna.goldenberg@utoronto.ca

## Abstract

When an image classifier makes a prediction, which parts of the image are relevant and why? We can rephrase this question to ask: which parts of the image, if they were not seen by the classifier, would most change its decision? Producing an answer requires marginalizing over images that could have been seen but weren't. We can sample plausible image in-fills by conditioning a generative model on the rest of the image. We then optimize to find the image regions that most change the classifier's decision after in-fill. Our approach contrasts with ad-hoc in-filling approaches, such as blurring or injecting noise, which generate inputs far from the data distribution, and ignore informative relationships between different parts of the image. Our method produces more compact and relevant saliency maps, with fewer artifacts compared to previous methods.

## 1 Introduction

The decisions of powerful image classifiers are difficult to interpret. Saliency maps are a tool for interpreting differentiable classifiers that, given a particular input example and output class, computes the sensitivity of the classification with respect to each input dimension. Fong & Vedaldi (2017) and Dabkowski & Gal (2017) cast saliency computation an optimization problem informally described by the following question: *which inputs, when replaced by an uninformative reference value, maximally change the classifier output?* Because these methods use heuristic reference values, e.g. blurred input (Fong & Vedaldi, 2017) or random colors (Dabkowski & Gal, 2017), they ignore the context of the surrounding pixels, often producing unnatural in-filled images (Figure 2). If we think of a saliency map as interrogating the neural network classifier, these approaches have to deal with a somewhat unusual question of how the classifier responds to images outside of its training distribution.

To encourage explanations that are consistent with the data distribution, we modify the question at hand: *which region, when replaced by plausible alternative values, would maximally change classifier output?* In this paper we provide a new model-agnostic framework for computing and visualizing feature importance of any differentiable classifier, based on variational Bernoulli dropout (Gal & Ghahramani, 2016). We marginalize out the masked region, conditioning the generative model on the non-masked parts of the image to sample counterfactual inputs that either change or preserve classifier behavior. By leveraging a powerful in-filling conditional generative model we produce saliency maps on ImageNet that identify relevant and concentrated pixels better than existing methods.

## 2 Related work

*Gradient-based approaches* (Simonyan et al., 2013; Springenberg et al., 2014; Zhang et al., 2016; Selvaraju et al., 2016) derive a saliency map for a given input example and class target by computing the gradient of the classifier output with respect to each component (e.g., pixel) of the input. The reliance on the local gradient information induces a bias due to gradient saturation or discontinuity in the DNN activations (Shrikumar et al., 2017). Adebayo et al. (2018) observed that some gradient-based saliency computation reflect an inductive bias due to the convolutional architecture, which is independent of the network parameter values.

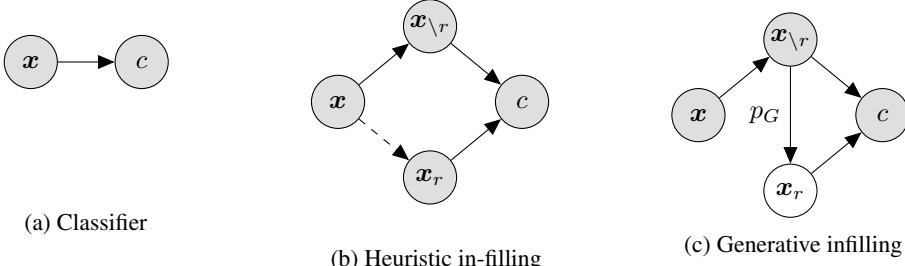

(a) Classifier

(b) Heuristic in-filling

(c) Generative infilling

Figure 1: **Graphical models**. (1a) $p_{\mathcal{M}}(c|\boldsymbol{x})$ is classifier whose behavior we wish to analyze. (1b) To explain its response to a particular input $\boldsymbol{x}$ we partition the input $\boldsymbol{x}$ into masked (unobserved) region $\boldsymbol{x}_r$ and their complement $\boldsymbol{x} = \boldsymbol{x}_r \cup \boldsymbol{x}_{\setminus r}$. Then we replace the $\boldsymbol{x}_r$ with uninformative reference value $\hat{\boldsymbol{x}}_r$ to test which region $\boldsymbol{x}_r$ is important for classifier's output $p_{\mathcal{M}}(c|\boldsymbol{x}_r, \boldsymbol{x}_{\setminus r})$. Heuristic in-filling (Fong & Vedaldi, 2017) computes $\hat{\boldsymbol{x}}_r$ ad-hoc such as image blur. This biases the explanation when samples $[\hat{\boldsymbol{x}}_r, \boldsymbol{x}_{\setminus r}]$ deviate from the data distribution $p(\boldsymbol{x}_r, \boldsymbol{x}_{\setminus r})$. (1c) We instead sample $\boldsymbol{x}_r$ efficiently from a conditional generative model $\boldsymbol{x}_r \sim p_G(\boldsymbol{x}_r|\boldsymbol{x}_{\setminus r})$ that respects the data distribution.

*Reference-based approaches* analyze the sensitivity of classifier outputs to the substitution of certain inputs/pixels with an uninformative reference value. Shrikumar et al. (2017) linearly approximates this change in classifier output using an algorithm resembling backpropagation. This method is efficient and addresses gradient discontinuity, but ignores nonlinear interactions between inputs. Chen et al. (2018) optimizes a variational bound on the mutual information between a subset of inputs and the target, using a variational family that sets input features outside the chosen subset to zero. In both cases, the choice of background value as reference limits applicability to simple image domains with static background like MNIST.

Zintgraf et al. (2017) computes the saliency of a pixel (or image patch) by treating it as unobserved and marginalizing it out, then measuring the change in classification outcome. This approach is similar in spirit to ours. The key difference is that where Zintgraf et al. (2017) iteratively execute this computation for each region, we leverage a variational Bernoulli distribution to efficiently search for optimal solution while encouraging sparsity. This reduces computational complexity and allows us to model the interaction between disjoint regions of input space.

Fong & Vedaldi (2017) computes saliency by optimizing the change in classifier outputs with respect to a perturbed input, expressed as the pixel-wise convex combination of the original input with a reference image. They offer three heuristics for choosing the reference: mean input pixel value (typically gray), Gaussian noise, and blurred input. Dabkowski & Gal (2017) amortize the cost of estimating these perturbations by training an auxiliary neural network.

## 3 PROPOSED METHOD

Dabkowski & Gal (2017) propose two objectives for computing the saliency map:

- Smallest Deletion Region (SDR) considers a saliency map as an answer to the question: *What is the smallest input region that could be removed and swapped with alternative reference values in order to minimize the classification score?*
- Smallest Supporting Region (SSR) instead poses the question: *What is the smallest input region that could substituted into a fixed reference input in order to maximize the classification score?*

Solving these optimization problems (which we formalize below) involves a search over input masks, and necessitates *reference values* to be substituted inside (SDR) or outside (SSR) the masked region. These values were previously chosen heuristically, e.g., mean pixel value per channel. We instead consider inputs inside (SDR) or outside (SSR) the masked region as unobserved variables to be marginalized efficiently by sampling from a strong conditional generative model[1]. We describe our approach for an image application where the input comprises pixels, but our method is more broadly applicable to any domain where the classifier is differentiable.

---

[1] Whereas (Zintgraf et al., 2017) iteratively marginalize single patches conditioned on their surroundings, we model a more expressive conditional distribution considering the joint interaction of non-contiguous pixels.

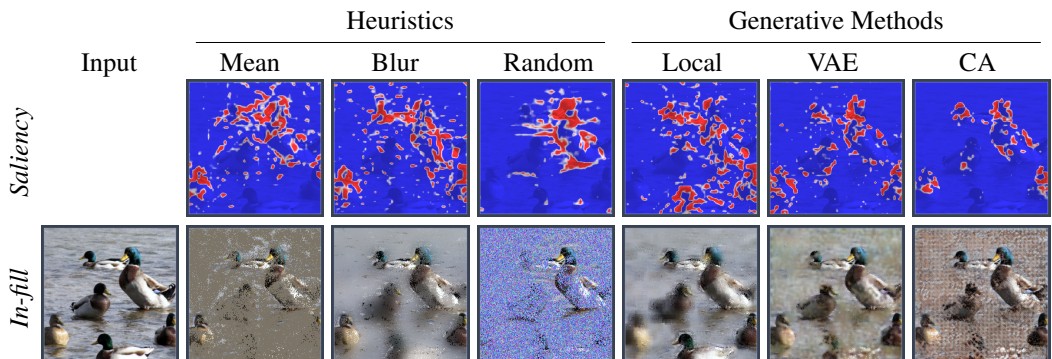

Figure 2: **Computed saliency for a variety of in-filling techniques**. The classifier predicts the correct label, "drake". Each saliency map (top row) results from maximizing in-class confidence of mixing a minimal region (red) of the original image with some reference image in the complementary (blue) region. The resulting mixture (bottom row) is fed to the classifier. We compare 6 methods for computing the reference, 3 heuristics and 3 generative models. We argue that strong generative models—e.g., Contextual Attention GAN (CA) (Yu et al., 2018)—ameliorate in-fill artifacts, making explanations more plausible under the data distribution.

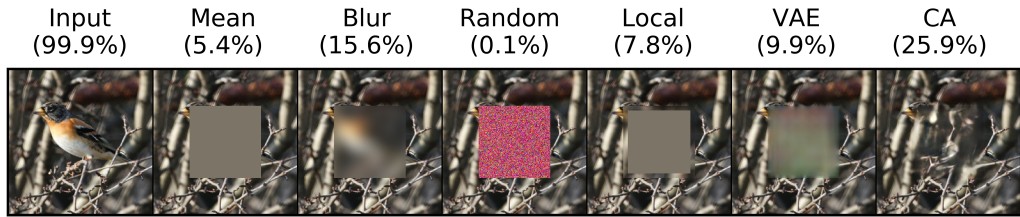

Figure 3: **Visualization of reference value infilling methods under centered mask**. The ResNet output probability of the correct class is shown (as a percentage) for each imputed image.

Consider an input image $\boldsymbol{x}$ comprising $U$ pixels, a class $c$, and a classifier with output distribution $p_{\mathcal{M}}(c|\boldsymbol{x})$. Denote by $r$ a subset of the input pixels that implies a partition of the input $\boldsymbol{x} = \boldsymbol{x}_r \cup \boldsymbol{x}_{\backslash r}$. We refer to $r$ as a region, although it may be disjoint. We are interested in the classifier output when $\boldsymbol{x}_r$ are unobserved, which can be expressed by marginalization as

$$p_{\mathcal{M}}(c|\boldsymbol{x}_{\backslash r}) = \mathbb{E}_{\boldsymbol{x}_r \sim p(\boldsymbol{x}_r|\boldsymbol{x}_{\backslash r})} \left[ p_{\mathcal{M}}(c|\boldsymbol{x}_{\backslash r}, \boldsymbol{x}_r) \right]. \tag{1}$$

We then approximate $p(\boldsymbol{x}_r|\boldsymbol{x}_{\backslash r})$ by some generative model with distribution $p_G(\boldsymbol{x}_r|\boldsymbol{x}_{\backslash r})$ (specific implementations are discussed in section 4.1). Then given a binary mask[2] $\boldsymbol{z} \in \{0, 1\}^U$ and the original image $\boldsymbol{x}$, we define an infilling function[3] $\phi$ as a convex mixture of the input and reference with binary weights,

$$\phi(\boldsymbol{x}, \boldsymbol{z}) = \boldsymbol{z} \odot \boldsymbol{x} + (\boldsymbol{1} - \boldsymbol{z}) \odot \hat{\boldsymbol{x}} \text{ where } \hat{\boldsymbol{x}} \sim p_G(\hat{\boldsymbol{x}}|\boldsymbol{x}_{\boldsymbol{z}=\boldsymbol{0}}). \tag{2}$$

### 3.1 OBJECTIVE FUNCTIONS

The classification score function $s_{\mathcal{M}}(c)$ represents a score of classifier confidence on class $c$; in our experiments we use log-odds:

$$s_{\mathcal{M}}(c|\boldsymbol{x}) = \log p_{\mathcal{M}}(c|\boldsymbol{x}) - \log(1 - p_{\mathcal{M}}(c|\boldsymbol{x})). \tag{3}$$

SDR seeks a mask $\boldsymbol{z}$ yielding low classification score when a small number of reference pixels are mixed into the mask regions. Without loss of generality[4], we can specify a parameterized distribution

---

[2] $\boldsymbol{z}_u = 0$ means the $u$-th pixel of $\boldsymbol{x}$ is dropped out. The remaining image is $\boldsymbol{x}_{\boldsymbol{z}=\boldsymbol{0}}$.

[3] The infilling function is stochastic due to randomness in $\hat{x}$.

[4] We can search for a single mask $\boldsymbol{z}'$ using a point mass distribution $q_{\boldsymbol{z}'}(\boldsymbol{z}) = \delta(\boldsymbol{z} = \boldsymbol{z}')$

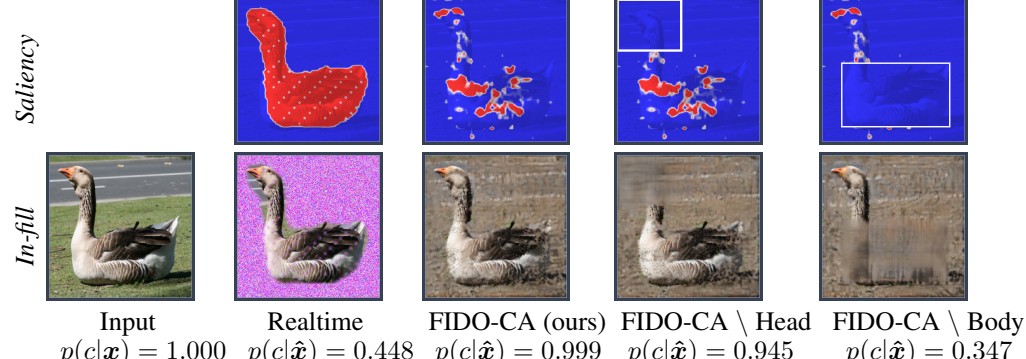

|  | Input | Realtime | FIDO-CA (ours) | FIDO-CA \ Head | FIDO-CA \ Body |
|---|---|---|---|---|---|
|  | $p(c|\boldsymbol{x}) = 1.000$ | $p(c|\hat{\boldsymbol{x}}) = 0.448$ | $p(c|\hat{\boldsymbol{x}}) = 0.999$ | $p(c|\hat{\boldsymbol{x}}) = 0.945$ | $p(c|\hat{\boldsymbol{x}}) = 0.347$ |

Figure 4: **Classifier confidence of infilled images.** Given an input, FIDO-CA finds a minimal pixel region that preserves the classifier score following in-fill by CA-GAN (Yu et al., 2018). Dabkowski & Gal (2017) (Realtime) assigns saliency coarsely around the central object, and the heuristic infill reduces the classifier score. We mask further regions (head and body) of the FIDO-CA saliency map by hand, and observe a drop in the infilled classifier score. The label for this image is "goose".

over masks $q_\theta(\boldsymbol{z})$ and optimize its parameters. The SDR problem is a minimization w.r.t $\theta$ of

$$L_{SDR}(\theta) = \mathbb{E}_{q_\theta(\boldsymbol{z})} \left[ s_{\mathcal{M}}(c|\phi(\boldsymbol{x}, \boldsymbol{z})) + \lambda \|\boldsymbol{z}\|_1 \right]. \tag{4}$$

On the other hand, SSR aims to find a masked region that maximizes classification score while penalizing the size of the mask. For sign consistency with the previous problem, we express this as a minimization w.r.t $\theta$ of

$$L_{SSR}(\theta) = \mathbb{E}_{q_\theta(\boldsymbol{z})} \left[ -s_{\mathcal{M}}(c|\phi(\boldsymbol{x}, \boldsymbol{z})) + \lambda \|\mathbf{1} - \boldsymbol{z}\|_1 \right]. \tag{5}$$

Naively searching over all possible $\boldsymbol{z}$ is exponentially costly in the number of pixels $U$. Therefore we specify $q_\theta(\boldsymbol{z})$ as a factorized Bernoulli:

$$q_\theta(\boldsymbol{z}) = \prod_{u=1}^{U} q_{\theta_u}(z_u) = \prod_{u=1}^{U} \mathrm{Bern}(z_u|\theta_u). \tag{6}$$

This corresponds to applying Bernoulli dropout (Srivastava et al., 2014) to the input pixels and optimizing the per-pixel dropout rate. $\theta$ is our saliency map since it has the same dimensionality as the input and provides a probability of each pixel being marginalized (SDR) or retained (SSR) prior to classification. We call our method FIDO because it uses a strong generative model (see section 4.1) to Fill-In the DropOut region.

To optimize the $\theta$ through the discrete random mask $\boldsymbol{z}$, we follow Gal et al. (2017) in computing biased gradients via the Concrete distribution (Maddison et al., 2017; Jang et al., 2017); we use temperature 0.1. We initialize all our dropout rates $\theta$ to 0.5 since we find it increases the convergence speed and avoids trivial solutions. We optimize using Adam (Kingma & Ba, 2014) with learning rate 0.05 and linearly decay the learning rate for 300 batches in all our experiments. Our PyTorch implementation takes about one minute on a single GPU to finish one image.

## 3.2 COMPARISON TO FONG & VEDALDI (2017)

Fong & Vedaldi (2017) compute saliency by directly optimizing the continuous mask $\boldsymbol{z} \in [0, 1]^U$ under the SDR objective, with $\hat{\boldsymbol{x}}$ chosen heuristically; we call this approach Black Box Meaningful Perturbations (BBMP). We instead optimize the parameters of a Bernoulli dropout distribution $q_\theta(\boldsymbol{z})$, which enables us to sample reference values $\hat{\boldsymbol{x}}$ from a learned generative model. Our method uses mini-batches of samples $\boldsymbol{z} \sim q_\theta(\boldsymbol{z})$ to efficiently explore the huge space of binary masks and obtain uncertainty estimates, whereas BBMP is limited to a local search around the current point estimate of the mask $\boldsymbol{z}$. See Figure 5 for a pseudo code comparison. In Appendix A.1 we investigate how the choice of algorithm affects the resulting saliency maps.

To avoid unnatural artifacts in $\phi(\boldsymbol{x}, \boldsymbol{z})$, (Fong & Vedaldi, 2017) and (Dabkowski & Gal, 2017) additionally included two forms of regularization: upsampling and total variation penalization.

| **Algorithm 1:** BBMP (Fong & Vedaldi, 2017) | **Algorithm 2:** FIDO (Ours) |
|---|---|
| **Input** image $x$, classifier score $s_{\mathcal{M}}$, sparsity hyperparameter $\lambda$, objective $L \in \{L_{SSR}, L_{SDR}\}$ | **Input** image $x$, classifier score $s_{\mathcal{M}}$, sparsity hyperparameter $\lambda$, objective $L \in \{L_{SSR}, L_{SDR}\}$, generative model $p_G$ |
| Initialize single mask $z \in [0,1]^d$ | Initialize dropout rate $\boldsymbol{\theta} \in [0,1]^d$ |
| **while** loss $L$ is not converged **do** | **while** loss $L$ is not converged **do** |
| $\quad$ Clip $z$ to $[0,1]$ | $\quad$ Sample minibatch $z \in \{0,1\}^d \sim \text{Bern}(\boldsymbol{\theta})$ |
| $\quad$ Compute reference image $\hat{x}$ heuristically | $\quad$ Sample reference image $\hat{x} \sim p_G(\hat{x}|z,x)$ |
| $\quad$ Compute in-fill | $\quad$ Compute in-fill |
| $\quad \phi(x, z) = x \cdot z + \hat{x} \cdot (1 - z)$ | $\quad \phi(x, z) = x \cdot z + \hat{x} \cdot (1 - z)$ |
| $\quad$ With $\phi$, compute $L$ by Equation 4 or 5 | $\quad$ With $\phi$, compute $L$ by Equation 4 or 5 |
| $\quad$ Update $z$ with $\nabla_z L$ | $\quad$ Update $\boldsymbol{\theta}$ with $\nabla_\theta L$ (Maddison et al., 2017) |
| **end while** | **end while** |
| Return $z$ as per-feature saliency map | Return $\boldsymbol{\theta}$ as per-feature saliency map |

Figure 5: **Pseudo code comparison**. Differences between the approaches are shown in blue.

Upsampling is used to optimize a coarser $\theta$ (e.g. $56 \times 56$ pixels), which is upsampled to the full dimensionality (e.g. $224 \times 224$) using bilinear interpolation. Total variation penalty smoothes $\theta$ by a $\ell_2$ regularization penalty between spatially adjacent $\theta_u$. To avoid losing too much signal from regularization, we use upsampling size 56 and total variation as 0.01 unless otherwise mentioned. We examine the individual effects of these regularization terms in Appendices A.2 and A.4, respectively.

## 4 EXPERIMENTS

We first evaluate the various infilling strategies and objective functions for FIDO. We then compare explanations under several classifier architectures. In section 4.5 we show that FIDO saliency maps outperform BBMP (Fong & Vedaldi, 2017) in a successive pixel removal task where pixels are in-filled by a generative model (instead of set to the heuristic value). FIDO also outperforms the method from (Dabkowski & Gal, 2017) on the so-called Saliency Metric on ImageNet. Appendices A.1–A.6 provide further analysis, including consistency and the effects of additional regularization.

### 4.1 INFILLING METHODS

We describe several methods for producing the reference value $\hat{x}$. The heuristics do not depend on $z$ and are from the literature. The proposed generative approaches, which produce $\hat{x}$ by conditioning on the non-masked inputs $x_{z=0}$, are novel to saliency computation.

**Heuristics:** **Mean** sets each pixel of $\hat{x}$ according to its per-channel mean across the training data. **Blur** generates $\hat{x}$ by blurring $x$ with Gaussian kernel ($\sigma = 10$) (Fong & Vedaldi, 2017). **Random** samples $\hat{x}$ from independent per-pixel per-channel uniform color with Gaussians ($\sigma = 0.2$).

**Generative Models:** **Local** computes $\hat{x}$ as the average value of the surrounding non-dropped-out pixels $x_{z=0}$ (we use a $15 \times 15$ window). **VAE** is an image completion Variational Autoencoder (Iizuka et al., 2017). Using the predictive mean of the decoder network worked better than sampling. **CA** is the Contextual Attention GAN (Yu et al., 2018); we use the authors' pre-trained model.

Figure 3 compares these methods with a centered mask. The heuristic in-fills appear far from the distribution of natural images. This is ameliorated by using a strong generative model like CA, which in-fills texture consistent with the surroundings. See Appendix A.5 for a quantitative comparison.

### 4.2 COMPARING THE SDR AND SSR OBJECTIVE FUNCTIONS

Here we examine the choice of objective function between $L_{SDR}$ and $L_{SSR}$; see Figure 6. We observed more artifacts in the $L_{SDR}$ saliency maps, especially when a weak in-filling method (Mean) is used. We suspect this unsatisfactory behavior is due to the relative ease of optimizing $L_{SDR}$. There are many degrees of freedom in input space that can increase the probability of any of the 999 classes besides $c$; this property is exploited when creating adversarial examples (Szegedy et al., 2013). Since

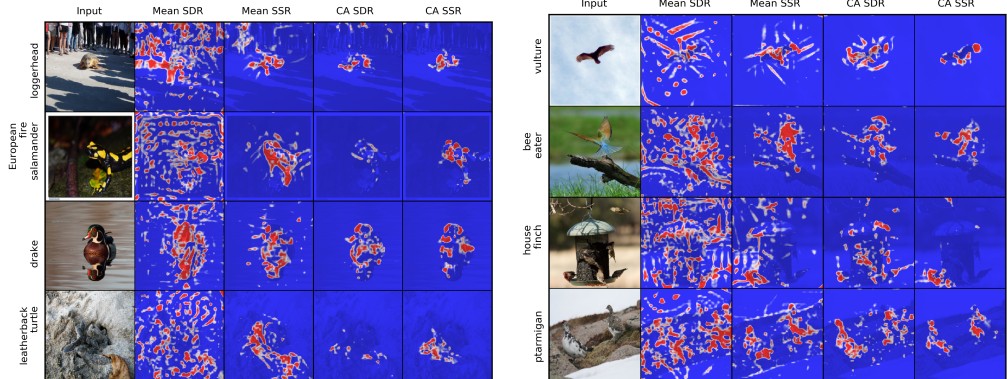

Figure 6: **Choice of objective between** $L_{SDR}$ **and** $L_{SSR}$. The classifier (ResNet) gives correct predictions for all the images. We show the $L_{SDR}$ and $L_{SSR}$ saliency maps under 2 infilling methods: Mean and CA. Here the red means important and blue means non-important. We find that $L_{SDR}$ is more susceptible to artifacts in the resulting saliency maps than $L_{SSR}$.

it is more difficult to infill unobserved pixels that increase the probability of a particular class $c$, we believe $L_{SSR}$ encourages FIDO to find explanations more consistent with the classifier's training distribution. It is also possible that background texture is easier for a conditional generative model to fit. To mitigate the effect of artifacts, we use $L_{SSR}$ for the remaining experiments.

## 4.3 COMPARING INFILLING METHODS

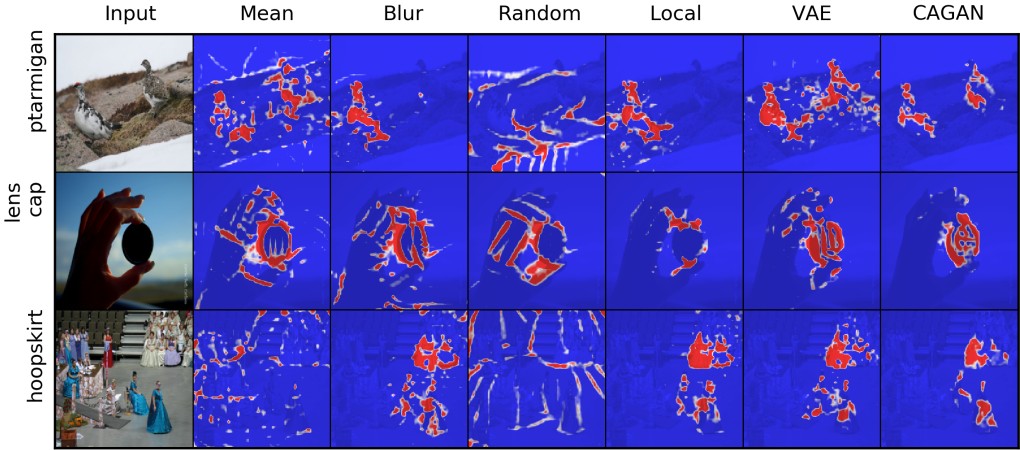

Figure 7: **Comparison of saliency map under different infilling methods** by FIDO SSR using ResNet. Heuristics baselines (Mean, Blur and Random) tend to produce more artifacts, while generative approaches (Local, VAE, CA) produce more focused explanations on the targets.

Here we demonstrate the merits of using strong generative model that produces substantially fewer artifacts and a more concentrated saliency map. In Figure 7 we generate saliency maps of different infilling techniques by interpreting ResNet using $L_{SSR}$ with sparsity penalty $\lambda = 10^{-3}$. We observed a susceptibility of the heuristic in-filling methods (Mean, Blur, Random) to artifacts in the resulting saliency maps, which may fool edge filters in the low level of the network. The use of generative in-filling (Local, VAE, CA) tends to mitigate this effect; we believe they encourage in-filled images to lie closer to the natural image manifold. To quantify the artifacts in the saliency maps by a proxy: the proportion of the MAP configuration ($\theta > 0.5$) that lies outside of the ground truth bounding box. FIDO-CA produces the fewest artifacts by this metric (Figure 8).

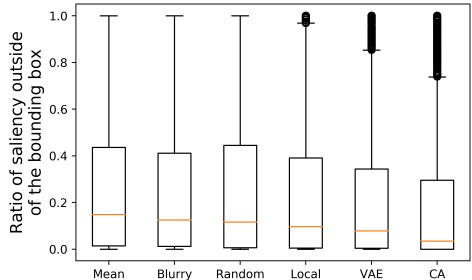

Figure 8: **Proportion of saliency map outside bounding box.** Different in-filing methods evaluating ResNet trained on ImageNet, $1,971$ images. The lower the better.

## 4.4 INTERPRETING VARIOUS CLASSIFIER ARCHITECTURES

We use FIDO-CA to compute saliency of the same image under three classifier architectures: AlexNet, VGG and ResNet; see Figure 9. Each architecture correctly classified all the examples. We observed a qualitative difference in the how the classifiers prioritize different input regions (according to the saliency maps). For example in the last image, we can see AlexNet focuses more on the body region of the bird, while Vgg and ResNet focus more on the head features.

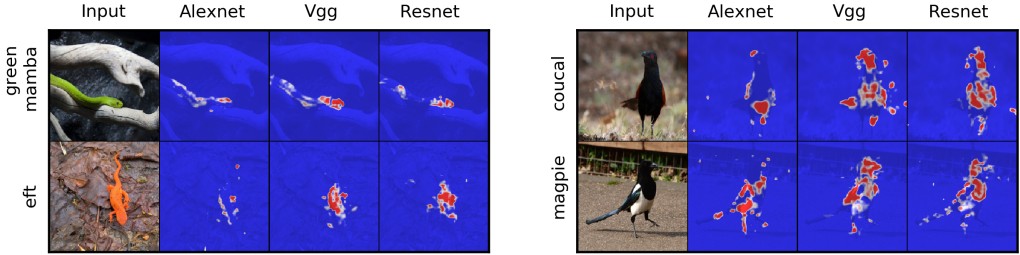

Figure 9: **Comparison of saliency maps for several classifier architectures**. We compare 3 networks: AlexNet, Vgg and ResNet using FIDO-CA with $\lambda = 10^{-3}$)

## 4.5 QUANTITATIVE EVALUATION

We follow Fong & Vedaldi (2017) and Shrikumar et al. (2017) in measuring the classifier's sensitivity to successively altering pixels in order of their saliency scores. Intuitively, the "best" saliency map should compactly identify relevant pixels, so that the predictions are changed with a minimum number of altered pixels. Whereas previous works flipped salient pixel values or set them to zero, we note that this moves the classifier inputs out of distribution. We instead dropout pixels in saliency order and infill their values with our strongest generative model, CA-GAN. To make the log-odds score suppression comparable between images, we normalize per-image by the final log-odds suppression score (all pixels infilled). In Figure 10 we evaluate on ResNet and carry out our scoring procedure on $1,533$ randomly-selected correctly-predicted ImageNet validation images, and report the number of pixels required to reduce the normalized log-odds score by a given percent. We evaluate FIDO under various in-filling strategies as well as BBMP with Blur and Random in-filling strategies. We put both algorithms on equal footing by using $\lambda = 1e-3$ for FIDO and BBMP (see Section A.1 for further comparisons). We find that strong generative infilling (VAE and CA) yields more parsimonious saliency maps, which is consistent with our qualitative comparisons. FIDO-CA can achieve a given normalized log-odds score suppression using fewer pixels than competing methods. While FIDO-CA may be better adapted to evaluation using CA-GAN, we note that other generative in-filling approaches (FIDO-Local and FIDO-VAE) still out-perform heuristic in-filling when evaluated with CA-CAN.

We compare our algorithm to several strong baselines on two established metrics. We first evaluate whether the FIDO saliency map can solve weakly supervised localization (WSL) (Dabkowski & Gal, 2017). After thresholding the saliency map $\theta$ above 0.5, we compute the smallest bounding box

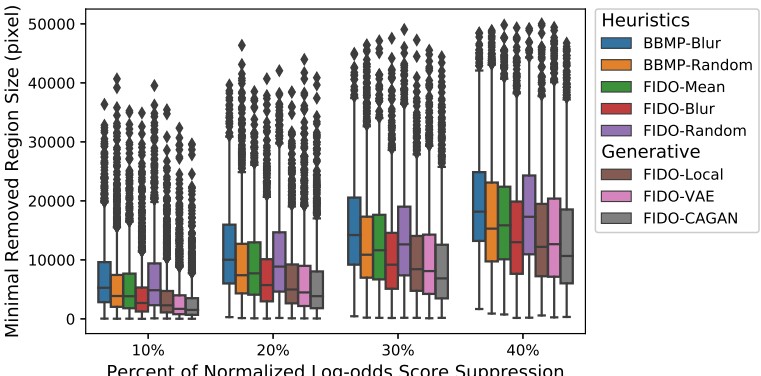

Figure 10: **Number of salient pixels required to change normalized classification score.** Pixels are sorted by saliency score and successively replaced with CA-GAN in-filled values. We select $\lambda = 1e-3$ for BBMP and FIDO. The lower the better.

| | Max | Center | Grad | Deconv | GradCAM | Realtime | FIDO | | | | | | BBox |
| --- | --- | --- | --- | --- | --- | --- | --- | --- | --- | --- | --- | --- | --- |
| | | | | | | | Mean | Blur | Random | Local | VAE | CA | |
| WSL | 59.7% | 46.9% | 52.1% | 49.4% | 41.9% | 47.8% | 53.1% | 51.6% | 50.8% | 46.7% | 50.2% | 57.0% | 0.0% |
| SM | 1.029 | 0.414 | 0.577 | 0.627 | 0.433 | 0.076 | 0.836 | 0.766 | 0.728 | 0.543 | 0.433 | **-0.021** | 0.392 |

Table 1: Weakly Supervised Localization (WSL) error and Saliency Metric (SM). All the methods evaluating under ResNet on 50k validation sets of ImageNet. For both metrics the lower the better.

containing all salient pixels. This prediction is "correct" if it has intersection-over-union (IoU) ratio over $0.5$ with any of the ground truth bounding boxes. Using FIDO with various infilling methods, we report the average error rate across all $50,000$ validation images in Table 1. We evaluate the authors' pre-trained model of Dabkowski & Gal (2017)[5], denoted as "Realtime" in the results. We also include five baselines: Max (entire input as the bounding box), Center (centered bounding box occupying half the image), Grad (Simonyan et al., 2013), Deconvnet (Springenberg et al., 2014), and GradCAM (Selvaraju et al., 2016). We follow the procedure of mean thresholding in Fong & Vedaldi (2017): we normalize the heatmap between $0$ and $1$ and binarize by threshold $\theta = \alpha\mu_i$ where $\mu_i$ is the average heatmap for image $i$. Then we take the smallest bounding box that encompasses all the binarized heatmap. We search $\alpha$ between $0$ to $5$ with $0.2$ step size on a holdout set to get minimun WSL error. The best $\alpha$ are $1.2$, $2$ and $1$ respectively.

FIDO-CA frugally assigns saliency to contextually important pixels while preserving classifier confidence (Figure 4), so we do not necessarily expect our saliency maps to correlate with the typically large human-labeled bounding boxes. The reliance on human-labeled bounding boxes makes WSL suboptimal for evaluating saliency maps, so we evaluate the so-called Saliency Metric proposed by Dabkowski & Gal (2017), which eschews the human labeled bounding boxes. The smallest bounding box $A$ is computed as before. The image is then cropped using this bounding box and upscaling to its original size. The Saliency Metric is $\log max(\text{Area}(A), 0.05) - \log p(c|\text{CropAndUpscale}(\boldsymbol{x}, A))$, the log ratio between the bounding box area and the in-class classifier probability after upscaling. This metric represents the information concentration about the label within the bounded region. From the superior performance of FIDO-CA we conclude that a strong generative model regularizes explanations towards the natural image manifold and finds concentrated region of features relevant to the classifier's prediction.

---

[5] We use the authors' PyTorch pre-trained model https://github.com/PiotrDabkowski/pytorch-saliency.

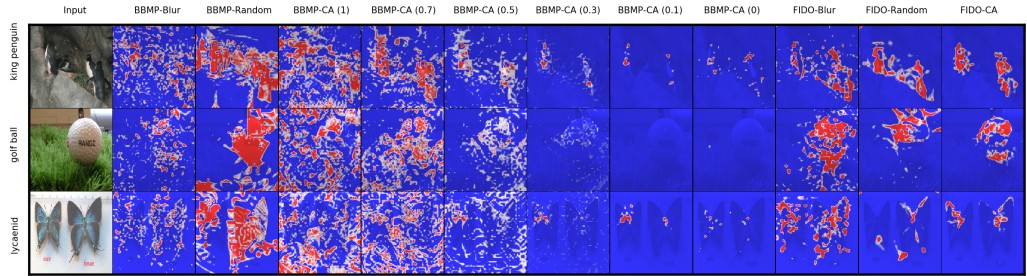

Figure 11: **Examples from the ablation study.** We show how each of our two innovations, FIDO and generative infilling, improve from previous methods that adopts BBMP with hueristics infilling (e.g. Blur and Random). Specifically, we compare with a new variant BBMP-CA that uses strong generative in-filling CA-GAN via thresholding the continous masks: we test a variety of decreasing thresholds. We find both FIDO (searching over Bernoulli masks) and generative in-filling (CAGAN) are needed to produce compact saliency maps (the right-most column) that retain class information. See Appendix B for more qualitative examples and in section A.7 for quantitative results.

### 4.6 ABLATION STUDY

Can existing algorithms be improved by adding an in-filling generative model without modeling a discrete distribution over per-feature masks? And does filling in the dropped-out region suffice without an expressive generative model? We carried out a ablation study that suggests no on both counts. We compare FIDO-CA to a BBMP variant that uses CA-GAN infilling (called BBMP-CA); we also evaluate FIDO with heuristic infilling (FIDO-Blur, FIDO-Random). Because the continuous mask of BBMP does not naturally partition the features into observed/unobserved, BBMP-CA first thresholds the masked region $r = \mathbb{I}(\boldsymbol{z} > \tau)$ before generating the reference $\phi(\boldsymbol{x}_r, \boldsymbol{x}_{\setminus r})$ with a sample from CA-GAN. We sweep the value of $\tau$ as $1, 0.7, 0.5, 0.3, 0.1$ and $0$. We find BBMP-CA is brittle with respect to its threshold value, producing either too spread-out or stringent saliency maps (Figure 11). We observed that FIDO-Blur and FIDO-Random produce more concentrated saliency map than their BBMP counterpart with less artifacts, while FIDO-CA produces the most concentrated region on the target with fewest artifacts. Each of these baselines were evaluated on the two quantitative metrics (Appendix A.7); BBMP-CA considerably underperformed relative to FIDO-CA.

## 5 SCOPE AND LIMITATIONS

Because the classifier behavior is ill-defined for out-of-distribution inputs, any explanation that relies on out-of-distribution feature values is unsatisfactory. By modeling the input distribution via an expressive generative model, we can encourage explanations that rely on counterfactual inputs close to the natural manifold. However, our performance is then upper-bounded by the ability of the generative model to capture the conditional input density. Fortunately, this bound will improve alongside future improvements in generative modeling.

## 6 CONCLUSION

We proposed FIDO, a new framework for explaining differentiable classifiers that uses adaptive Bernoulli dropout with strong generative in-filling to combine the best properties of recently proposed methods (Fong & Vedaldi, 2017; Dabkowski & Gal, 2017; Zintgraf et al., 2017). We compute saliency by marginalizing over plausible alternative inputs, revealing concentrated pixel areas that preserve label information. By quantitative comparisons we find the FIDO saliency map provides more parsimonious explanations than existing methods. FIDO provides novel but relevant explanations for the classifier in question by highlighting contextual information relevant to the prediction and consistent with the training distribution. We released the code in PyTorch at `https://github.com/zzzace2000/FIDO-saliency`.

ACKNOWLEDGMENTS

We thank Yu et al. (2018) for releasing their code and pretrained model CA-GAN that makes this work possible. We also thank Jesse Bettencourt and David Madras for their helpful suggestions.

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

# A  FURTHER ANALYSIS

## A.1  COMPARISONS OF BBMP AND FIDO

Here we compare FIDO with two previously proposed methods, BBMP with Blur in-filling strategy (Fong & Vedaldi, 2017) and BBMP with Random in-filling strategy (Dabkowski & Gal, 2017). One potential concern in qualitatively comparing these methods is that each method might have a different sensitivity to the sparsity parameter $\lambda$. Subjectively, we observe that BBMP requires roughly 5 times higher sparsity penalty $\lambda$ to get visually comparable saliency maps. In our comparisons we sweep $\lambda$ over a reasonable range for each method and show the resulting sequence of increasingly sparse saliency maps (Figure 12). We use $\lambda = 5e{-}4, 1e{-}3, 2e{-}3, 5e{-}3$.

We observe that all methods are prone to artifacts in the low $\lambda$ regime, so the appropriate selection of this value is clearly important. Interestingly, BBMP Blur and Random respectively find artifacts with different quality: small patches and pixels for Blur and structure off-object lines for Random. FIDO with CA is arguably the best saliency map, producing fewer artifacts and concentrating saliency on small regions for the images.

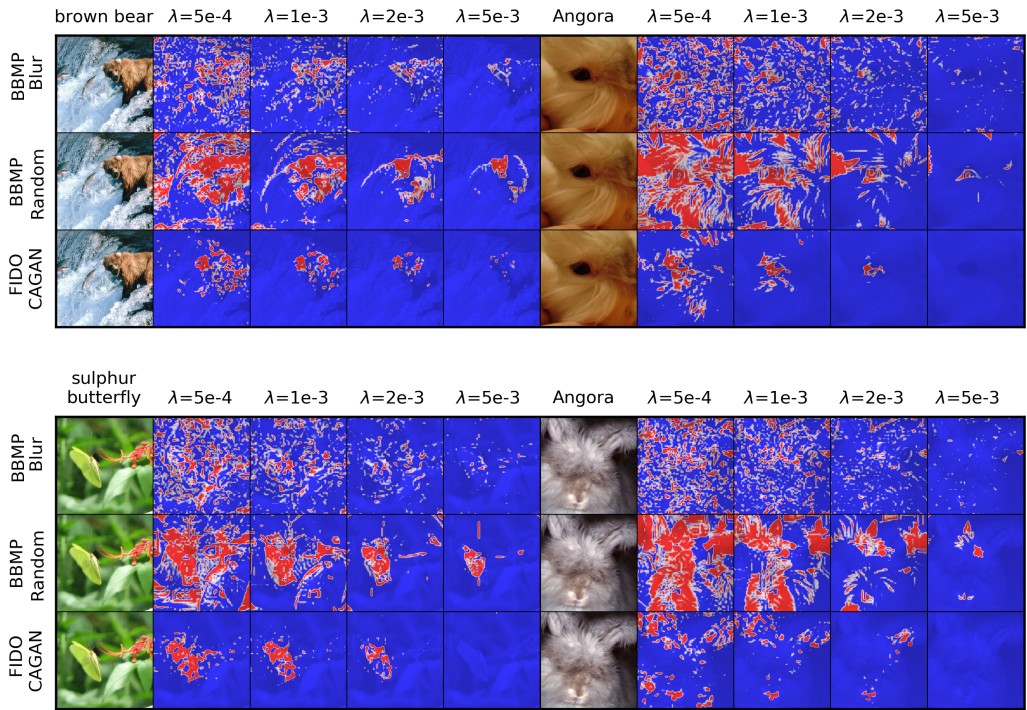

Figure 12: **BBMP vs FIDO saliency map** by increasing the sparsity penalty $\lambda$ value from left to right. We compare with BBMP under Blur and Random, and FIDO under CA in-filling strategies. Note that here BBMP and FIDO methods use different $\lambda$. We show the heuristics produce more artifacts (BBMP Blur) or produces weird lines (BBMP Random) compared to our method (FIDO CAGAN).

## A.2  UPSAMPLING EFFECT

Here we examine the effect of learning a reduced dimensionality $\theta$ that upsampled to the full image size during optimization. We consider a variety of upsampling rates, and in a slight abuse of terminology we refer to the upsampling "size" as the square root of the dimensionality of $\theta$ before upsampling, so smaller size implies more upsampling. In Figure 13, we demonstrate two examples with different upsampling size under Mean and CA infilling methods with SSR objectives. The weaker infilling strategy Mean apparently requires stronger regularization to avoid artifacts compared to CA. Note that although CA produces much less artifacts compared to Mean, it still produces

some small artifacts outside of the objects which is unfavored. We then choose 56 for the rest of our experiments to balance between details and the removal of the artifacts.

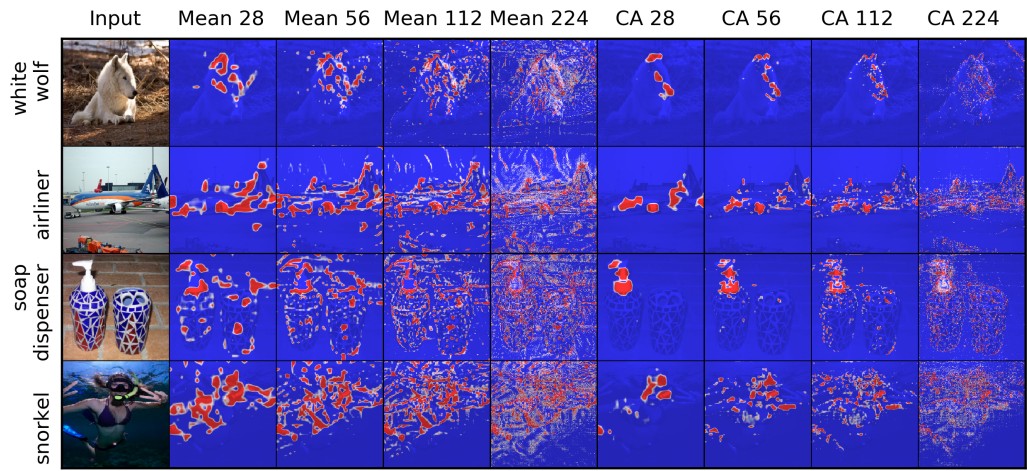

Figure 13: **Comparisons of upsampling effect** in Mean and CA infilling methods with no total variation penalty. We show the upsampling regularization removes the artifacts especially in the weaker infilling method Mean.

## A.3 STABILITY

To show the stability of our method, we test our method with different random seeds and observe if they are similar. In Figure 14, our method produces similar saliency map for 4 different random seeds.

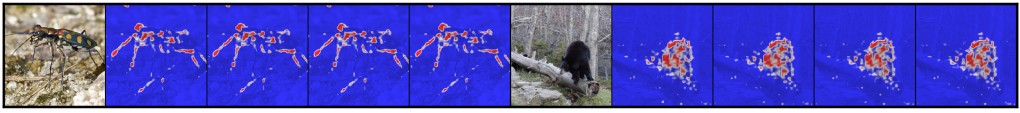

Figure 14: **Testing the stability of our method** with 4 different random seeds. They produce similar saliency maps. (Using CA infilling method with ResNet and $\lambda = 10^{-3}$)

## A.4 TOTAL VARIATION EFFECT

Here we test the effect of total variation prior regularization in Figure 15. We find the total variation can reduce the adversarial artifacts further, while risking losing signals when the total variation penalty is too strong.

## A.5 ANALYSIS OF GENERATIVE MODEL INFILLING

Here we quantitatively compare the in-filling strategies. The generative approaches (VAE and CA) perform visually sharper images than four other baselines. Since we expect this random removal should not remove the target information, we use the classification probability of the ResNet as our metric to measure how good the infilling method recover the target prediction. We quantitatively evaluate the probability for $1,000$ validation images in Figure 16. We find that VAE and CA consistently outperform other methods, having higher target probability. We also note that all the heuristic baselines (Mean, Blur, Random) perform much worse since the heuristic nature of these approaches, the images they generate are not likely under the distribution of natural images leading to the poor performance by the classifier.

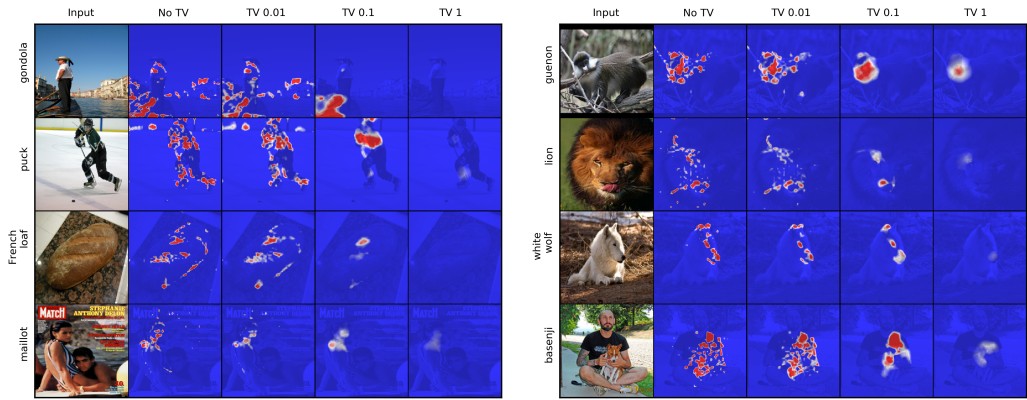

Figure 15: **Total Variation Regularization Effect**. We show the saliency maps of 4 increasing total variation (TV) regularization. We show that strong regularization risks removing signal.

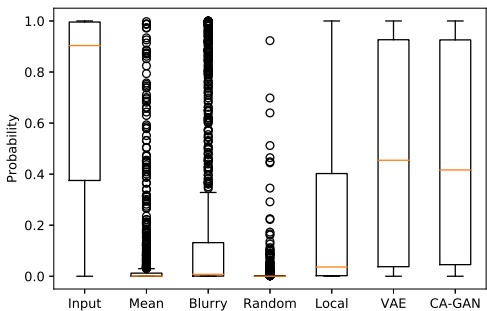

Figure 16: **Box plot of the classifier probability under different infilling with respect to random masked pixels** using ResNet under 1,000 images. We show that generative models (VAE and CA) performs much better in terms of classifier probability.

## A.6 BATCH SIZE EFFECTS

Figure 17 shows the effect of batch size on the saliency map. We found unsatisfactory results for batch size less than 4, which we attribute this to the high variance in the resulting gradient estimates.

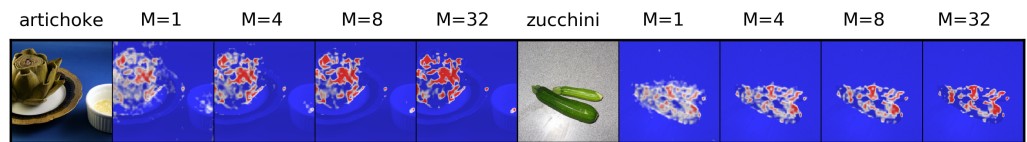

Figure 17: **Batch size effects of final saliency output**. We observed unsatisfactory results for batch size less than 4.

## A.7 ABLATION STUDY

We show the performance of BBMP-CA with various thresholds $\tau$ on both WSL and SM on subset of 1,000 images in Table 2. We also show more qaulitative examples in Figure 22. We find BBMP-CA is relatively brittle across different thresholds of $\tau$. Though with $\tau = 0.3$, the BBMP-CA perform

slightly better than BBMP and FIDO with heuristics infilling, it still performs substantially inferior to FIDO-CA. We also perform the flipping experiment in Figure 18 and show our FIDO-CA substantially outperforms BBMP-CA with varying different thresholds.

| | BBMP | | BBMP-CA | | | | | | | | FIDO | | | |
|---|---|---|---|---|---|---|---|---|---|---|---|---|---|---|
| | Blur | Random | 0 | 0.1 | 0.2 | 0.3 | 0.4 | 0.5 | 0.7 | 1 | Blur | Random | CA | BBox |
| WSL | 61.3% | 55.0% | 68.6% | 90.2% | 89.1% | 79.1% | 57.5% | 63.1% | 63.5% | 63.5% | 48.7% | 53.3% | 52.1% | 0.0% |
| SM | 1.097 | 0.912 | 0.722 | 1.426 | 1.272 | 0.535 | 0.595 | 1.103 | 1.123 | 1.117 | 0.763 | 0.784 | **-0.092** | 0.432 |

Table 2: Weakly Supervised Localization (WSL) error and Saliency Metric (SM) with comparisons on BBMP-CA with varying thresholds $\tau$. FIDO (various in-filling methods) evaluating ResNet on $1,000$ validation sets of ImageNet. For both metrics the lower the better.

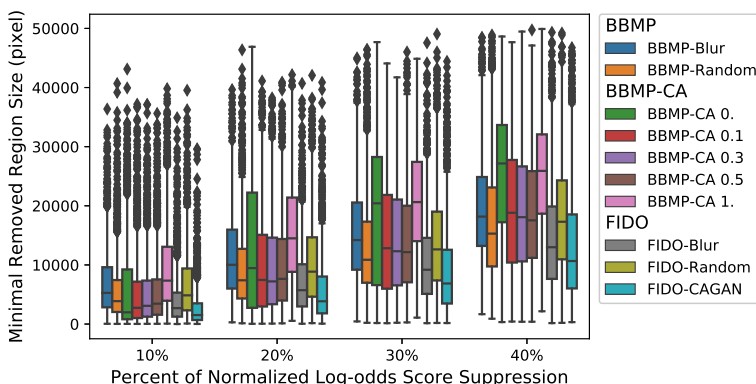

Figure 18: **Number of salient pixels required to change normalized classification score** with comparison to BBMP-CA across variety of thresholds. Pixels are sorted by saliency score and successively replaced with CA-GAN in-filled values. The lower the better.

## B MORE EXAMPLES

Figure 19 shows several more infilled counterfactual images, along with the counterfactuals produced by the method from Dabkowski & Gal (2017). More examples comparing the various FIDO infilling approaches can be found in Figure 20 and 21.

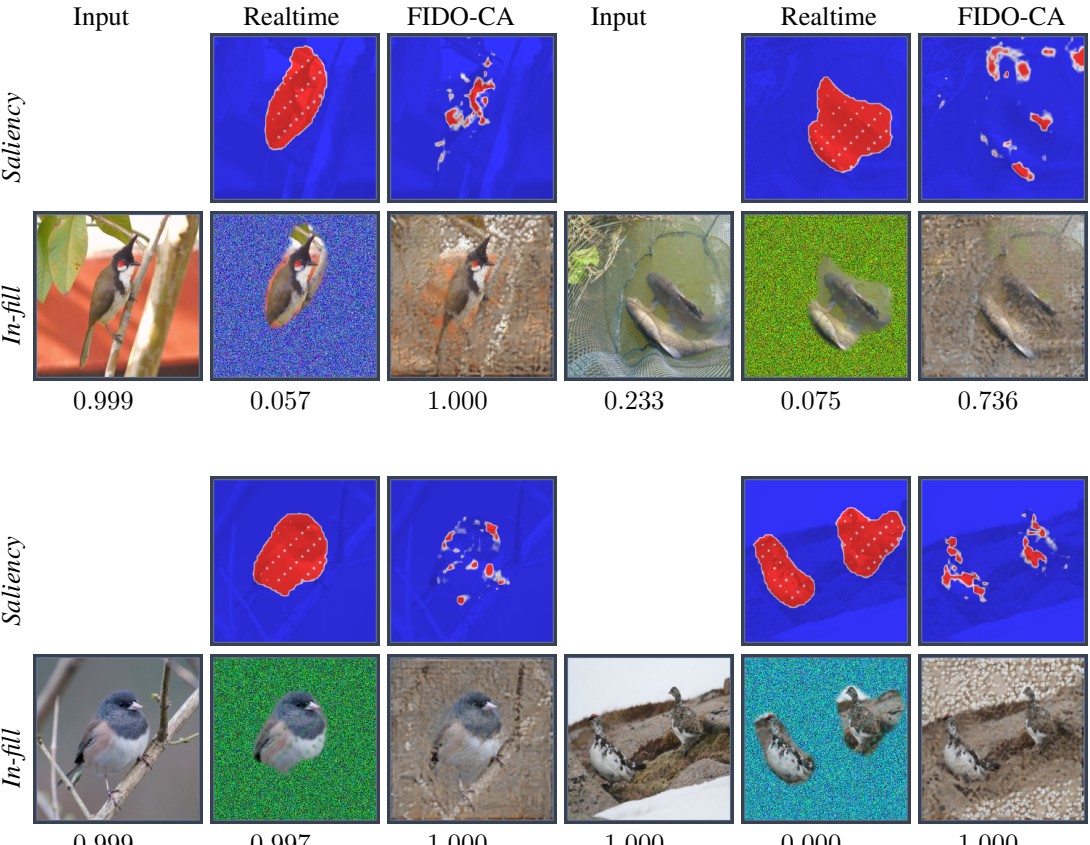

Figure 19: **More examples of classifier confidence on infilled images.** Realtime denotes the method of Dabkowski & Gal (2017); FIDO-CA is our method with CA-GAN infilling (Yu et al., 2018). Classifier confidence $p(c|\hat{x})$ is reported below the input and each infilled image. We hypothesize by that FIDO-CA is able to isolate compact pixel areas of contextual information. For example, in the upper right image pixels in the net region around the fish are highlighted; this context information is missing from the Realtime saliency map but are apparently relevant to the classifier's prediction. These 4 examples are bulbul, tench, junco, and ptarmigan respectively.

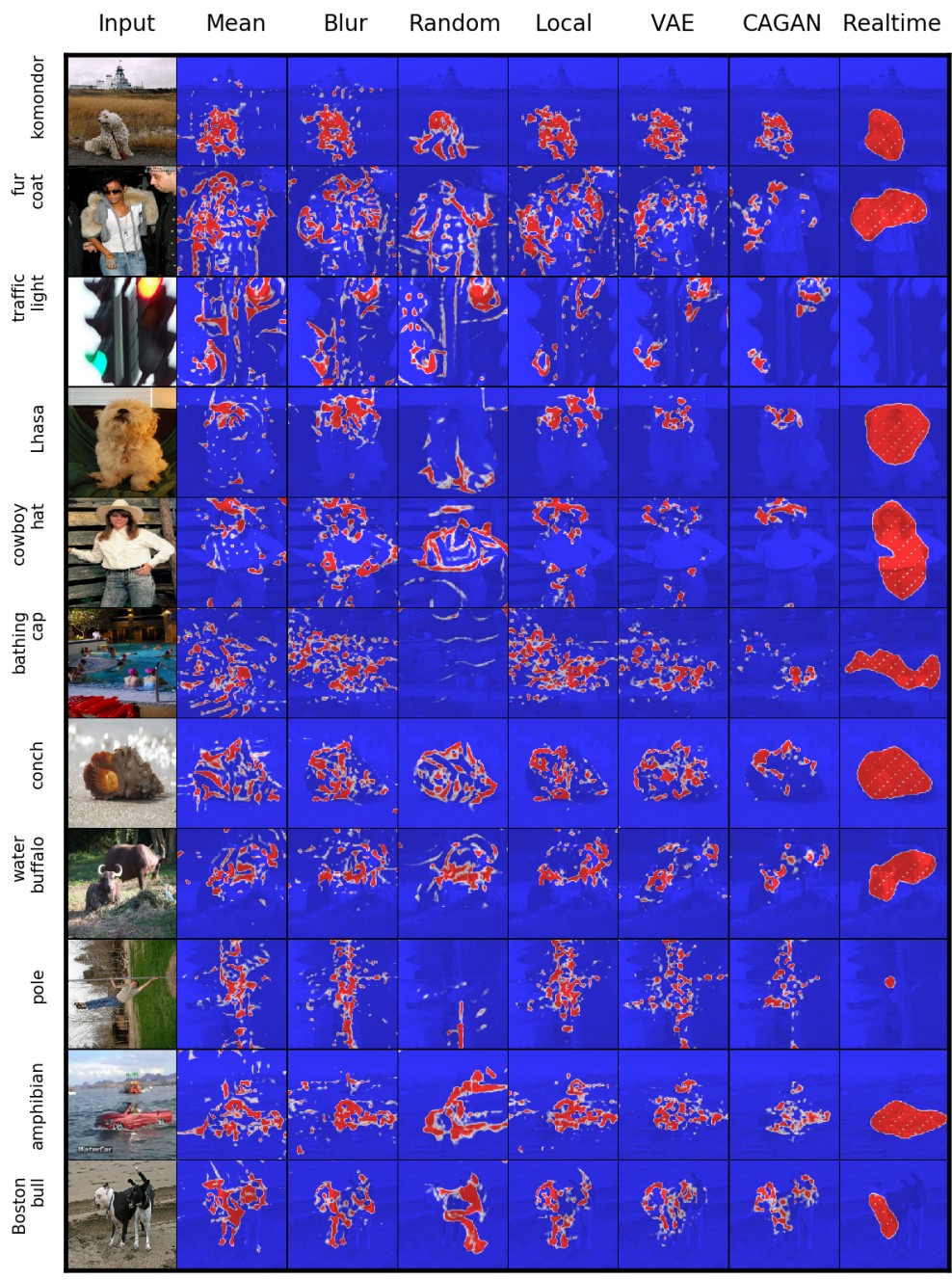

Figure 20: **Additional Saliency Maps** for FIDO under total variation 0.01 with a variety of in-filling methods. We include the method from Dabkowski & Gal (2017) in the right-most column.

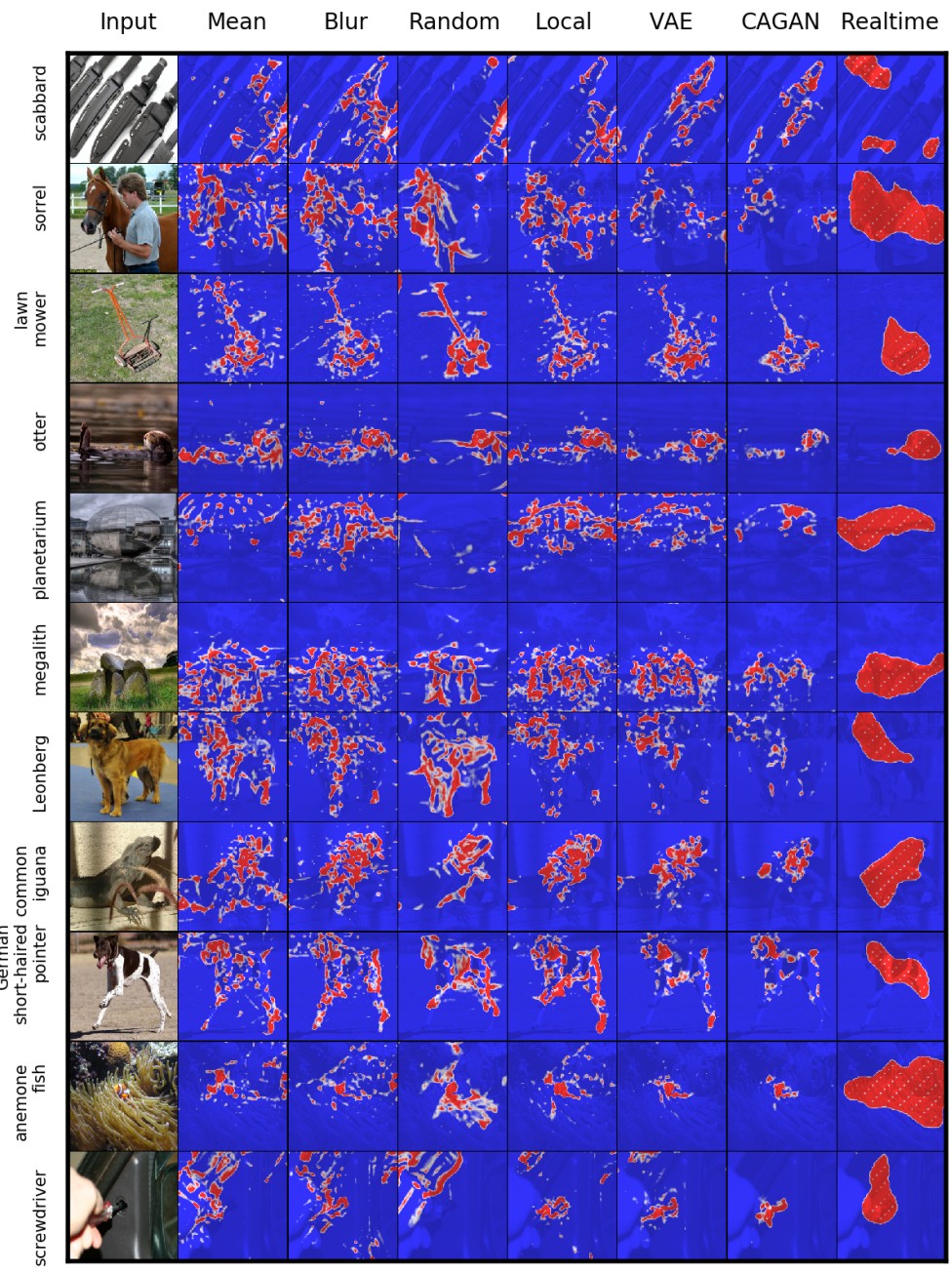

Figure 21: **Additional Saliency Maps** for FIDO under total variation $0.01$ with a variety of in-filling methods. We include the method from Dabkowski & Gal (2017) in the right-most column.

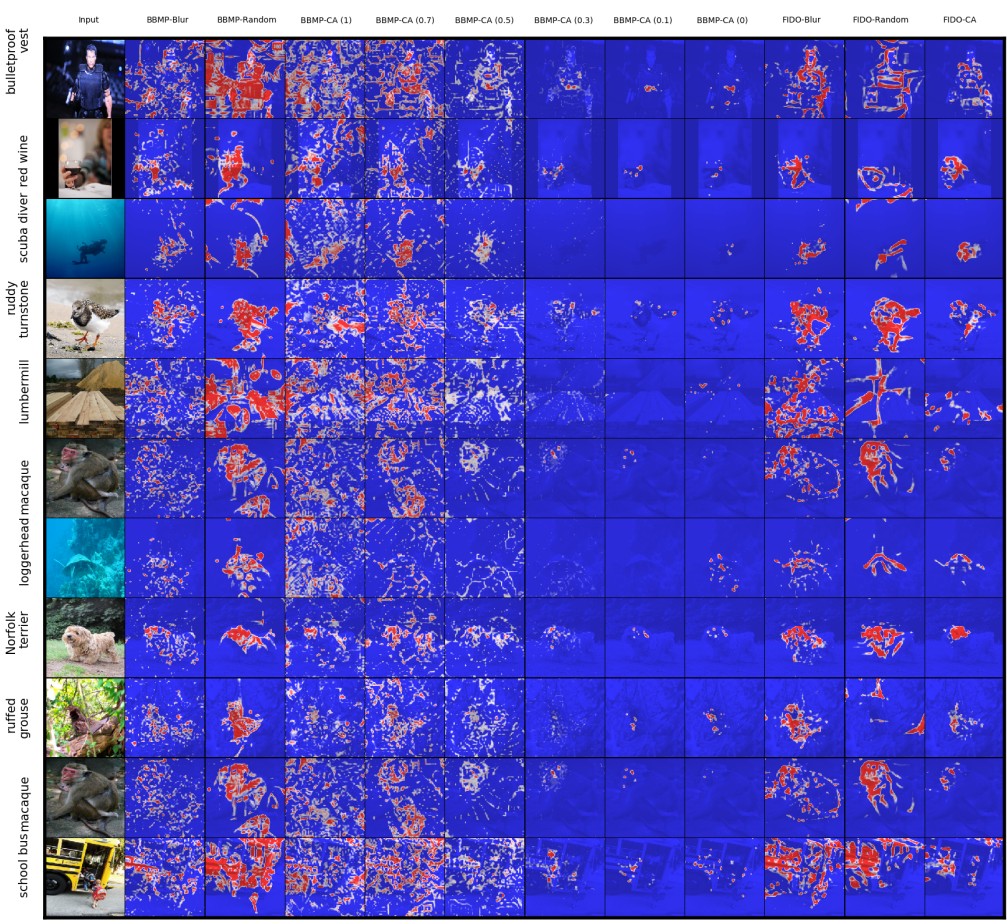

Figure 22: **Additional examples of ablation study.**

