# OpenReview forum: "Explaining Image Classifiers by Counterfactual Generation"
_ICLR.cc/2019/Conference_

### Official Review · AnonReviewer1 · 2018-10-21
**A paper with good motivation and results although the novelty and justifications are somewhat lacking.**

**Rating:** 5
**Confidence:** 5

**Review:**

Summary: This paper aims to find important regions to classify an image. The main algorithm, FIDO, is trained to find a saliency map based on SSR or SDR objective functions. The main novelty of this work is that it uses generative models to in-fill masked out regions by SSR or SDR. As such, compared to existing algorithms, FIDO can synthesize more realistic samples to evaluate.

I like the motivation of this paper since existing algorithms have clear limitations, i.e., using out-of-distribution samples. This issue can be addressed by using a generative network as described in this paper.

However, I think this approach yields another limitation: the performance of the algorithm is bound by the generative network. For example, let’s assume that a head region is important to classify birds. Also assume that the proposed algorithm somehow predicts a mask for the head region during training. If the generative network synthesizes a realistic bird from the mask, then the proposed algorithm will learn that the head region is a supporting region of SSR. In the other case, however, the rendered bird is often not realistic and classified incorrectly. Then, the algorithm will seek for other regions. As a result, the proposed method interprets a classifier network conditioned on the generative network parameters. Authors did not discuss these issues importantly in the paper.

Although the approach has its own limitation, I still believe that the overall direction of the paper is reasonable. It is because I agree that using a generative network to in-fill images to address the motivation of this paper is the best option we have at this current moment. In addition, authors report satisfactory amount of experimental results to support their claim.

Quality: The paper is well written and easy to follow.

Clarify: The explanation of the approach and experiments are clear. Since the method is simple, it also seems that it is easy to reproduce their results.

Originality: Authors apply off-the-shelf algorithms to improve the performance of a known problem. Therefore, I think there is no technical originality except that authors found a reasonable combination of existing algorithms and a problem.

Significance: The paper has a good motivation and deals with an important problem. Experimental results show improvements. Overall, the paper has some amount of impact in this field.

Pros and Cons are discussed above. As a summary,
Pros:
+ Good motivation.
+ Experiments show qualitative and quantitative improvements.

Cons:
- Lack of technical novelty and justification of the approach.

---

> ### Author Response · Authors · 2018-11-17
> **Thank you for your thorough assessment of our work**
>
> We thank you very much for your thorough review and for acknowledging that our proposed use of generative models is a sensible approach and its significance over the field. We believe that this conceptual contribution will help to progress interpretability beyond current limitations when most of the methods are still based on out-of-distribution inputs.
>
> We respectfully disagree with your assessment of the paper as lacking technical originality, as our method is not combining two off-the-shelf methods. Integrating a powerful generative model into current saliency algorithms efficiently is non-trivial (and to our knowledge has not been done, whereas your review might suggest such an algorithm already exists). The difficulty of combining existing generative models with existing saliency algorithms BBMP is evidenced by a new ablation study in the revision (section 4.6). It shows that naive combination of BBMP and CA perform much worse than our method FIDO. By parameterizing a distribution over dropped-out input features, FIDO provides a principled and efficient way to integrate over sensible counterfactual inputs to determine the relevant input features to a rendered prediction.
>
> You have raised an important point about the performance of this framework being upper bounded by the capacity of the in-filling generative model. We believe the reader will benefit from a discussion of this limitation, which we include in the revision (section 5). It is true that the ability to explain the classifier c(y|x) that learns p(y|x) is now somehow tied to the ability of the generative model g(x) to fit p(x). However, we strongly believe optimization-based saliency strategies that ignore or over-simplify p(x) (as existing methods do) are fundamentally misspecifying the counterfactual “what if” question that will yield an explanation. Moreover this upper bound on performance will increase in the future as generative models improve.

---

### Official Review · AnonReviewer2 · 2018-11-03
**Very well-written paper that introduces to important innovations to the problem of interpreting black-box NNs**

**Rating:** 7
**Confidence:** 3

**Review:**

The paper is aimed at answering the following question: "for model M, given an instance input and a predicted label, what parts of the input are most relevant for making the M choose the predicted label?".
This is by far not the first paper aimed at answering this question, but it makes important innovations to the best of my knowledge. The most important one is proposing a stronger approach to the counterfactual question "had this part of the input been different, what would have been the output?". Because the input can be different in many ways, an important question is addressing in what specific way would it have been different.

Specifically in the domain of images, most models assume a blurring or simple local in-painting approach: "if this patch were just a blurry average, what would have been the output?". However, ss the current paper correctly points out, blurring or other simple in-painting methods leads to an image which is outside the manifold of natural images and outside the domain of the training set. This can lead to biased or inaccurate results.

The paper therefore propose two innovations on top of existing methods, most closely building on work by Fong & Vedaldi (2017):
(1) Optimizing an inference network for discovering image regions which are most informative
(2) Using a GAN to in-paint the proposed regions, leading to a much more natural image and a more meaningful counterfactual question.

The presentation is crisp, especially the pseudo-code in Figure 5. In addition, the paper includes several well-executed experiments assessing the contributions of different design choices on different metrics and making careful comparisons with several recent methods addressing the same problem.

Specific comments:

1. In sec. 4.5, the comparison is not entirely fair because FIDO was already trained with CA-GAN, and therefore might be better adapted for it.
2. Related to the point above: could one train BBMP with a CA-GAN in-painting model?
3. I would have liked to see an ablation experiment where either one of the two innovations presented in this paper is missing.


Minor:
1. In eq. (2), wouldn't it be more accurate to denote it as \phi(x,z,\hat{x}) ?
2. I would like to know the true labels for all the examples presented in the paper.

---

> ### Author Response · Authors · 2018-11-17
> **Thank you for your thorough assessment of our work**
>
> We thank you for your thorough assessment of our work. You have concisely summarized the key contribution, and we agree with your explanation of how including a generative model of the input space allows FIDO to ask a more meaningful counterfactual question than existing approaches.
>
> In response to your specific comments:
> 1. We totally agree that the ideal evaluation here would sample from the true conditional infilling distribution, so the fact that we instead sample from CA-GAN is a limitation and might lead to a preferable performance on FIDO-CA. However we still observe a win that using other generative models (Local and VAE) over heuristics (Mean, Blur and Random). This suggests that generative infilling can still identify more relevant pixels corresponding to the classifier. We include this limitation in our revision.
> 2. Extending BBMP for use with a generative in-filler is not natural since it optimizes over continuous masks in [0, 1] rather than parameters of discrete masks in {0, 1} so the mask does not partition features into observed/unobserved. But we implemented an attempt at this in the revision and describe the result below.
> 3. We believe the reader will also benefit from this ablation study. In section 4.6 of the revision we investigate whether BBMP could be improved by using the CA-GAN to do infill. We threshold the BBMP masks then in-fill with CA-GAN. We find that this approach---called BBMP-CA---remains susceptible to artifacts in the resulting saliency maps and is brittle w.r.t its threshold value. BBMP-CA performs worse on the quantitative metrics than FIDO-CA, and about on par with FIDO-Blur and FIDO-Random, which do not use expressive generative models. Therefore we believe that one must model a discrete distribution over masks (not a point estimate like BBMP) in order to leverage the expressivity of an in-filling generative model.
>
> In response to your minor comments:
> 1. It is true that \phi has an indirect dependence on \hat x. But since \hat x is dependent also on x and z as a random variable drawn from the generative models, we think \phi(x, z) is still valid in this case. We make note of the stochasticity of \phi in the revision.
> 2. We include all the true labels in the revision.

---

### Official Review · AnonReviewer3 · 2018-11-04
**Unclear improvement over state-of-the-art saliency maps extractors**

**Rating:** 5
**Confidence:** 4

**Review:**

This paper introduces a new saliency map extractor to visualize which input features are relevant for the deep neural network to recognize objects in the image. The proposed saliency map extractor searches over a big space of potentially relevant image features and in-fills the irrelevant image regions using generative models.

The algorithmic machinery in the paper is poorly justified, as it is presented as a series of steps without providing much intuition why these steps are useful (especially compared to previous works). Also, I would like to know how this paper compares to Fan et al. "Adversarial localization network" (NIPS workshop, 2017), which has not been cited and it proposes similar ideas.

Also, the results are not convincing. Only one previous work (among many) has been compared with the proposed algorithm, and the qualitative examples are not enlightening showing the advantages of the introduced saliency map extractor. What are the new insights into the functioning of deep networks that were gained from the proposed saliency map extractor?

In summary, it is unclear to me if there is any novelty in the approach (missing references, lack of motivation of the algorithm) and if the results show any improvement over previous works (only one previous work has been compared and the qualitative examples do not show anything particularly interesting).

---

> ### Author Response · Authors · 2018-11-17
> **Thank you for your helpful review, clarification provided in the comment**
>
> We thank you very much for your effort in assessing our work, and for pointing us to the workshop paper on weakly localized supervision by Fan et al. 2017. We suspect that any lack of clarity about our method---its motivation, novelty, and improvement relative to baselines---is due to a misunderstanding about the scope of our paper and its key contribution. We hope to clarify this here and explain why Fan et al. 2017 is not a suitable baseline.
>
> Our goal is to explain the prediction produced by a differentiable classifier (that has been previously trained and whose weights are frozen) on a new test input x’. We formulate this as a search for features of x’ that change the classifier prediction significantly when they are marginalized out in a probabilistic framework. By contrast BBMP also searches over masks in continuous [0,1] to a point estimate and infills with heuristics (rather than marginalizing). This makes this method susceptible to artifacts in the computed saliency since it produces an explanation that relies on out-of-distribution (o.o.d.) inputs, where the classifier behavior isn’t well specified. Our key technical differences with BBMP (see blue text in figure 5) are firstly using Bernoulli distribution over masks, and secondly use an expressive generative model for efficient marginalization. These are novel to our knowledge, and neither of these differences are workable alone using existing algorithms; we add a new ablation study in the revision to emphasize this.
>
> Meanwhile, Fan et al. seek to solve weakly supervised localization (WSL) of objects in images using adversarial training. The goal of WSL is to locate the object, not to explain a pre-trained classifier; Fan et al 2017 include a classifier in their model, but this classifier’s weights are trained by their algorithm. We do not train the classifier, since we are trying to explain its predictions. Also, Fan et al use background infilling rather than a strong generative model. It is possible that a generative infilling model could be trained jointly with a classifier for improved WSL relative to Fan et al, but that is orthogonal to the goal and scope of our work.
>
> Despite common usage within saliency map papers, WSL is not a fully satisfactory evaluation for saliency map algorithms. Firstly (related to the above point) saliency map algorithms attempt to explain a known classifier rather than predict object localizations. For example, if the classifier ignores the object and classifies based on contextual information, then the correct saliency map should score poorly on WSL because it will also ignore the object. Nevertheless for completeness we evaluated FIDO on this task. There are some other shortcomings of WSL as a saliency metric that we can discuss if you are curious. All of this is motivation for the “saliency metric” proposed by Dabowski and Gal 2017, which we also evaluate. In the revision we also compare against a larger class of baseline models (Grad, DeconvNet, GradCAM) along both metrics, which we hope addresses your concern about how our model compares with other methods from the literature.

---

> > ### Comment · AnonReviewer3 · 2018-12-14
> > **Provide comparison**
> >
> > Fan et al. is used in saliency prediction and seems to achieve good accuracy as reported in other papers:
> > https://openreview.net/forum?id=BJxbYoC9FQ

---

> > > ### Author Response · Authors · 2018-12-18
> > > **Fan et al. is orthogonal to our work**
> > >
> > > As we mentioned above, Fan et al. is orthogonal to our work. We highly recommend you to reread our manuscript to understand the scope of our work.

---

### Author Response · Authors · 2018-11-17
**Revision of the paper**

We thank each of the reviewers for their thoughtful comments, which have helped us to improve the paper in the latest revision. We made the following changes:
- We include a new ablation study to help the reader better understand the importance of each technical contribution. The specific goal is to understand whether BBMP, the method most closely related to ours, could be improved with CA-GAN infilling, and find that BBMP+CA-GAN substantially underperforms relative FIDO+CA-GAN. This points toward that the FIDO framework is necessary to leverage expressive generative models in the interpretation of classifiers. We discuss this experiment in further detail below in the response to AnonReviewer2.
- For our quantitative evaluations (weakly supervised localization and Dabowski & Gal 2017’s “saliency metric”) we evaluate three additional baseline models. These are Gradient-based class saliency (Simonyan et al 2013), DeconvNet (Springenberg et al 2014), and GradCAM (Selvajaru et al 2016).
- We expand our discussion to better describe how the FIDO framework depends on the capacity of the generative model.
- We confirm our original findings with increased statistical confidence by evaluating over the entire validation set (50k images). We note there is a discrepency of WSL performances with what Dabowski and Gal 2017 reported. We try to resolve it by communicating with the authors but unfortunatelly they are unable to provide neither the evaluation code nor the original model they use. However for completeness we still compare with this model.
- Ground truth labels are now displayed beside the images for the qualitative comparison of saliency maps.
- In the supplement we show how batch size of mask samples M affects the saliency computed by FIDO-CA. Performance degrades with small batch size (< 4).
- We include additional qualitative examples in the supplementary.

We summarize our key contributions:
- We propose a novel framework, called FIDO, for explaining classifier decisions that efficiently search for explanations that respect the distribution of input data by generative model.
- We show that incorporating strong generative models reduces artifacts substantially and provides more relevent pixels of explanation. This addresses the common shortcomings of existing methods that uses out-of-distribution (o.o.d.) data that leads to increasing artifacts, as shown in our experiment.
- We quantitatively show the generative models perform better than heuristics infill on two widely-used evaluation methods. We also extensively compare with the recent literature.
- We also show SDR (used by Fong & Vedaldi, 2017) is prone to a much higher degree of artifact compared to SSR qualitatively.

The individual concerns of each reviewer will be addressed in the comments below. Please let us know if you have additional comments, and if there are particular revisions that would increase your assessment of our paper.

---

> ### Comment · AnonReviewer1 · 2018-12-11
> **Thanks for the rebuttal...**
>
>
> The rebuttal addresses some of the issues...
> * Figure 11 now clearly shows that the proposed algorithm is not merely a combination of two existing approaches.
> * Authors mentioned and discussed the limitation of this approach in Section 5.
>
> After reading the revised paper, I have additional comments.
> It is known that a classification network can be fooled by a small amount of (adversarial) noise. It is also true that an image inpainting algorithm inevitably synthesizes some artifacts. Then, why do artifacts rendered by the inpainting algorithm not severely corrupt the task that tries to find good regions for classification? Is it just because artifacts are not adversarially generated? It is not necessary but it would be great if the paper discusses this aspect as well.

---

> > ### Author Response · Authors · 2018-12-11
> > **Thanks for reading the rebuttal**
> >
> > Good question. I think it's not just because it's not adversarially generated, since other heuristics infilling are also not trained to do so.
> >
> > I think in high dimensional datasets like images, the infilling has so much freedom to generate out-of-distribution inputs. Image inpainting algorithm explicitly train to restict the infilling to natural images, so it makes the infilling harder to find adversarial perturbations. [1] also has similar insights of using generative models that protects it from the adversarial attack. I also think it's also the reason for fewer artifacts of SSR than SDR since finding the evidence for 1 class have way less freedom than finding evidence for other 999 classes.
> >
> > Thank you for reading the rebuttal. We will include this discussion in the version later.
> >
> > [1] Defense-GAN: Protecting Classifiers Against Adversarial Attacks Using Generative Models
> > https://openreview.net/forum?id=BkJ3ibb0-

---

### Meta-Review · Area_Chair1 · 2018-12-14

**Confidence:** 4
**Recommendation:** Accept (Poster)

**Metareview:**

Important problem (explainable AI); sensible approach, one of the first to propose a method for the counter-factual question (if this part of the input were different, what would the network have predicted). Initially there were some concerns by the reviewers but after the author response and reviewer discussion, all three recommend acceptance (not all of them updated their final scores in the system).